# Freezing In with lepton flavored fermions

G. D'Ambrosio[1★], Shiuli Chatterjee[2†], Ranjan Laha[2,3‡] and Sudhir K Vempati[2∘]

**1** INFN-Sezione di Napoli, Complesso Universitario di Monte S. Angelo,
Via Cintia Edificio 6, 80126 Napoli, Italy
**2** Centre for High Energy Physics, Indian Institute of Science, Bangalore 560012, India
**3** Theoretical Physics Department, CERN, 1211 Geneva, Switzerland

★ gdambros@na.infn.it, † shiulic@iisc.ac.in, ‡ ranjanlaha@iisc.ac.in, ∘ vempati@iisc.ac.in

## Abstract

Dark, chiral fermions carrying lepton flavor quantum numbers are natural candidates for freeze-in. Small couplings with the Standard Model fermions of the order of lepton Yukawas are 'automatic' in the limit of Minimal Flavor Violation. In the absence of total lepton number violating interactions, particles with certain representations under the flavor group remain absolutely stable. For masses in the GeV-TeV range, the simplest model with three flavors, leads to signals at future direct detection experiments like DARWIN. Interestingly, freeze-in with a smaller flavor group such as $SU(2)$ is already being probed by XENON1T.

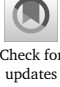

# 1   Introduction

One of the intriguing questions of particle dark matter (DM) is whether it is a single particle or comes in different varieties or flavors. In fact, in many BSM (Beyond Standard Model) models, it is not unusual to find a flavored DM. The sneutrino in MSSM [1] is one such popular candidate. Independent of specific BSM models, ideas of flavored DM have been in development for some time [2–8]. In another interesting paper, Batell *et al.* [9]. have shown that imposing minimal flavor violation (MFV) automatically ensures stability of the DM in the quark sector. In the leptonic sector however such a stability is not guaranteed [10].

In the present work, we focus on flavored DM and study its relevance to the freeze-in mechanism [11–13]. The conditions required for a successful production of the relic density through this mechanism are in contrast to the "WIMP" (Weakly Interacting Massive Particle) paradigm. The dark particle is never in thermal equilibrium, pushing its couplings with the Standard Model (SM) particles to quite low values, in some cases as low as $\sim \mathcal{O}(10^{-6})$. Modelling the Feebly Interacting Massive Particle (FIMP) DM typically involves two scenarios. The first scenario that is considered is called Ultraviolet (UV) freeze-in where the DM is produced at high temperatures through non-renormalizable interactions with the SM particles. In this case, the effective scale suppressing the non-renormalizable operators sets the couplings small. In the second case of the Infrared (IR) freeze-in, the relic density is sourced via small renormalizable couplings. Various techniques have been adopted to arrive at these two scenarios [14–23, 23–30]. In recent times, there have also been attempts towards a full effective theory for freeze-in DM [31].

In this work, we discuss flavored DM within the context of freeze-in mechanism. The DM is assumed to have non-trivial representations under the charged lepton flavor group of the SM. We show that this choice is apt for the freeze-in mechanism. The neutrino sector on the other hand, is far more model dependent and could also lead to freeze-in mechanisms for certain models (see for example, [32,33]). We also address the issue of stability for charged lepton flavored DM and show that in the absence of total lepton number violating interactions, they can be absolutely stable. Of the possible stable representations, chiral representations are interesting for freeze-in. In particular, those which have the same chiral representation as those of the SM leptons. We call them as *flavor mirrors*. This is in similar vein as models of mirror Standard Model, but restricting to only the flavor symmetry [34–36]. As is sort of well understood, in this the couplings of the dark fermions are determined by the Yukawa couplings. In particular, the lightest of them could couple with the electron Yukawa making it an interesting candidate for freeze-in. Furthermore, it can be shown that at the lowest order of the effective theory, at dimension 5, both the operators, the Higgs operator as well as the dipole operators are chirally suppressed. The Higgs portal and the magnetic dipole moment operators play the dominant role in setting the relic density whereas the dipole operators, especially the electric dipole moment, dominate in direct detection rates. In fact, for masses of $\mathcal{O}(1\text{-}100)$ GeV, we find that there would be signatures at the future direct detection experiments like LZ and DARWIN. For a smaller flavor group like the $SU(2)$, we find that there are already constraints from XENON1T.

The paper is organized as follows. In the next section, we address the issue of stability under lepton flavor groups. In section III, we focus on minimal chiral flavored DM and show that it satisfies the relic density and study its direct detection signatures. We also show the results for the smaller flavor group $SU(2)$. We conclude in section IV.

Table 1: Shown are the lowest dimension flavor representations. The columns are, respectively, representation of $\chi_L$ and $\chi_R$ under $G_{LF}$, its lepton number ($q_{LN}$), whether MFV requirement is essential to the stability argument, whether lepton number conservation (LNC) is imposed, whether $\chi$ is stable, and the lowest dimension operators allowed under MFV.

| $\chi_L$ | $\chi_R$ | $q_{LN}$ | MFV | LNC | Stable | Operators |
|---|---|---|---|---|---|---|
| (3,1) | (1,3) | 0 | | ✓ | ✓ | $(\bar{\chi}_L \sigma_{\mu\nu} Y_l \chi_R) B^{\mu\nu}, (\bar{\chi}_L \sigma_{\mu\nu}\gamma_5 Y_l \chi_R) B^{\mu\nu}, (\bar{\chi}_L Y_l \chi_R) H^\dagger H$ |
| (3,1) | (1,3) | -1 | ✓ | ✓ | ✓ | $(\bar{\chi}_L \sigma_{\mu\nu} Y_l \chi_R) B^{\mu\nu}, (\bar{\chi}_L \sigma_{\mu\nu}\gamma_5 Y_l \chi_R) B^{\mu\nu}, (\bar{\chi}_L Y_l \chi_R) H^\dagger H$ |
| (3,1) | (3,1) | -1 | ✓ | ✓ | ✓ | $(\bar{\chi}_L \sigma_{\mu\nu} \chi_R) B^{\mu\nu}, (\bar{\chi}_L \sigma_{\mu\nu}\gamma_5 \chi_R) B^{\mu\nu}, (\bar{\chi}_L \chi_R) H^\dagger H$ |
| (3,1) | (1,3) | 1 | ✓ | ✓ | | |
| (6,1) | (1,6) | 1 | ✓ | ✓ | ✓ | $(\bar{\chi}_L \sigma_{\mu\nu} Y_l^\dagger Y_l^\dagger \chi_R) B^{\mu\nu}, (\bar{\chi}_L \sigma_{\mu\nu}\gamma_5 Y_l^\dagger Y_l^\dagger \chi_R) B^{\mu\nu}, (\bar{\chi}_L \sigma_{\mu\nu} Y_l Y_l^\dagger \chi_R) H^\dagger H$ |
| (6,1) | (1,6) | -1 | ✓ | ✓ | | |
| (8,1) | (1,8) | 1,-1 | ✓ | ✓ | ✓ | $(\bar{\chi}_L \sigma_{\mu\nu} Y_l Y_l^\dagger \chi_R) B^{\mu\nu}, (\bar{\chi}_L \sigma_{\mu\nu}\gamma_5 Y_l Y_l^\dagger \chi_R) B^{\mu\nu}, (\bar{\chi}_L \sigma_{\mu\nu} Y_l Y_l^\dagger \chi_R) H^\dagger H$ |
| (8,1) | (1,8) | 0 | | ✓ | ✓ | $(\bar{\chi}_L \sigma_{\mu\nu} Y_l Y_l^\dagger \chi_R) B^{\mu\nu}, (\bar{\chi}_L \sigma_{\mu\nu}\gamma_5 Y_l Y_l^\dagger \chi_R) B^{\mu\nu}, (\bar{\chi}_L \sigma_{\mu\nu} Y_l Y_l^\dagger \chi_R) H^\dagger H$ |

## 2 Lepton Flavored Dark Matter and its Stability

In the absence of Yukawa couplings, the leptonic part of the SM Lagrangian has an additional global $SU(3)$ covering over the three generations of electron, muon and tau[1]. This symmetry is valid separately for the doublets $L_i$ ($i = 1, 2, 3$, generation index) and the singlets $E_i$ of the $SU(2)_L$ of the SM gauge group. The total flavor group in the charged lepton sector is given by:

$$G_{LF} \equiv SU(3)_L \otimes SU(3)_E. \tag{1}$$

The Yukawa terms break the flavor symmetry which becomes evident when $G_{SM}$ is broken via the Higgs mechanism: $L_Y = \bar{L} Y_l E H + h.c.$, where $Y_l$ is a $3 \times 3$ matrix in flavor/generational space and generation indices of the fields are suppressed.

The leptonic doublets and singlets transform non-trivially under the $G_{LF}$ as:

$$L \sim (3,1)_{G_{LF}}, \ E \sim (1,3)_{G_{LF}}. \tag{2}$$

The dark sector does not carry any of the SM gauge quantum numbers as it is assumed to be a SM singlet. On the other hand, it could come in three generations like the charged leptons in the SM. Its interactions with the SM, however, could either induce additional flavor violation or restrict to the violation present in $Y_l$. The latter falls under the paradigm of MFV. Formally, this is achieved by treating the Yukawa matrix as a (spurion) field that has a non-trivial transformation under $G_{LF}$, $Y_l \sim (3, \bar{3})_{G_{LF}}$, such that the Yukawa term now becomes invariant under $G_{LF}$. Then, models with additional flavored matter content can be studied as effective field theories (EFTs) under the MFV hypothesis where the Yukawa is a spurion and we only include operators that are invariant under $G_{LF}$, in addition to the SM gauge symmetry.

The MFV ansatz not only restricts the interactions of the DM particles with SM fermions but also has implications for the stability of the DM. It was shown that these restrictions can in and of themselves lead to the stability of specific representation of quark flavored DM [9]. We will follow stability arguments similar to refs. [9,10] assuming a lepton flavored, SM gauge singlet, Dirac fermion DM, $\chi$. In the following we will consider not just the triplet representations as mentioned above, but also other higher dimensional representations like sextets and octets.

---

[1]This is in addition to the individual lepton numbers of $U(1)_{L_e}, U(1)_{L_\mu}, U(1)_{L_\tau}$

The most generic operator that can cause the decay of DM $\chi$ is:

$$\mathcal{O}_{decay} = \chi \underbrace{L...}_{A} \underbrace{\bar{L}...}_{B} \underbrace{E...}_{C} \underbrace{\bar{E}...}_{D} \underbrace{Y_l...}_{E} \underbrace{Y_l^\dagger...}_{F} \mathcal{O}_{weak}, \tag{3}$$

with one $\chi$ field decaying into $A$ number of $L$ fields, $B$ number of $\bar{L}$ fields, $C$ number of $E$ fields, $D$ number of $\bar{E}$ fields, $E$ number of the spurion $Y_l$ and $F$ number of the spurion $Y_l^\dagger$. The operator $\mathcal{O}_{weak}$ accounts for other weak fields that render $\mathcal{O}_{decay}$ an SM gauge singlet.

In order for $\mathcal{O}_{decay}$ to be allowed under MFV, it must be a singlet under $G_{LF} \equiv SU(3)_L \otimes SU(3)_E$. The invariance of the operator $\mathcal{O}_{decay}$, under the various $SU(3)$ groups becomes evident in the $(p, q)$ notation [37] of each representation. Noting that a tensor product $(p, q)_i$ with $p$ factors of $\mathbf{3}_i$ and $q$ factors of $\bar{\mathbf{3}}_i$ is invariant under the corresponding $SU(3)_i$ if

$$(p - q)_i \bmod 3 = 0. \tag{4}$$

We denote the irreducible representation of $\chi$ under $G_{LF}$ as:

$$\chi \sim (n_L, m_L)_L \times (n_E, m_E)_E, \tag{5}$$

and get the following conditions for $\mathcal{O}_{decay}$ to be allowed:

$$\begin{aligned} SU(3)_L : \ & (A - B + E - F + n_L - m_L) \bmod 3 = 0, \\ SU(3)_E : \ & (C - D - E + F + n_E - m_E) \bmod 3 = 0. \end{aligned} \tag{6}$$

Adding these two equations gives:

$$(A - B + C - D + n_L - m_L + n_E - m_E) \bmod 3 = 0. \tag{7}$$

This condition depends on the number of SM fields appearing in $\mathcal{O}_{decay}$, in addition to $\chi$'s representation under $G_{LF}$, and therefore at best gives stability conditions dimension by dimension, and cannot predict absolute stability [10]. A special case of interest is the case where the interactions conserve total lepton number. Then for $O_{decay}$ to be allowed while conserving total lepton number gives a condition:

$$(A - B + C - D) = 0, \tag{8}$$

and subtracting this from eq. (7) leads to a condition dependent only on the DM flavor representation. Stating the condition for the most general decay operator $O_{decay}$ to be *forbidden* then gives us the stability condition:

$$(n_L - m_L + n_E - m_E) \bmod 3 \neq 0. \tag{9}$$

This condition automatically predicts the absolute stability of specific representations. These arguments can also be generalized for the case of a DM with a non-zero lepton number $q_{LN}$, in which case eq. (8) gets modified to:

$$(A - B + C - D + q_{LN}) = 0 \implies (A - B + C - D + q_{LN}) \bmod 3 = 0, \tag{10}$$

which taken together with eq. (7) gives the following condition for *stability*:

$$(n_L - m_L + n_E - m_E - q_{LN}) \bmod 3 \neq 0. \tag{11}$$

The stability condition thus depends on the lepton number $q_{LN}$ in addition to the flavor representation, eq. (5), of the DM, given all interactions conserve total lepton number.

We also note that a zero or rational, non-integer lepton number assignment can make a *fermionic* DM trivially stable, independent of any flavor structures, and can be argued from just the condition for an even number of fermions $(1 + A + B + C + D) \bmod 2 = 0$ and LNC $(q_{LN} + A - B + C - D) = 0 \implies (q_{LN} + A - B + C - D) \bmod 2 = 0$. These two together then give for DM stability:

$$(2A + 2C + q_{LN} + 1) \bmod 2 \neq 0 \implies (q_{LN} + 1) \bmod 2 \neq 0. \tag{12}$$

As long as this condition is satisfied, the DM is completely stable.

In Table 1, we list the lowest dimension representations of DM $\chi$ and their stabilities in accordance to eqs. (11) and (12). The first 2 columns give the flavor representations of the left and right chiralities of DM $\chi$, with the DM lepton number listed in the third column and in the sixth column we list a $\checkmark$ if such a DM is rendered stable by eq. (11) or eq. (12). In the fourth column, a $\checkmark$ means the MFV condition given by eqs. (6) has been implemented to investigate the stability, while if eq. (12) by itself implies stability of DM, the MFV column is left empty (then implementing MFV from eqs. (6) neither adds to nor spoils the stability argument). We have assumed in all the cases that the interactions conserve lepton number, as can be seen from the $\checkmark$'s in the fifth column. We see that in addition to the flavor representations, the sign as well as the magnitude of lepton number of the DM candidate dictate its stability. Finally, the sixth column lists the lowest dimension operators for DM interaction with SM for cases where the DM is stable.

We note that although a vector-like representation, third row of table 1, can be rendered stable, the interactions shown in last column are to leading order not suppressed by the charged lepton Yukawa matrices and do not lead to the small couplings. Hence, the representations of interest to us that naturally lead to a freeze-in production are the ones that are chiral as well as absolutely stable. Finally a note is in order. Imposition of lepton number and/or baryon number conservation leading to a stable DM is found in many BSM models too. For example, the R-parity conservation in MSSM [38] the KK Parity in UED models [39], and fractional baryon number conservation in RS models [40].

# 3 Minimal Flavored Chiral Dark Matter

We choose a chiral fermionic DM ($\chi$), and in particular the simplest flavor representation,

$$\chi_L \sim (3,1)_{G_{LF}}, \qquad \chi_R \sim (1,3)_{G_{LF}}, \qquad \chi = \chi_L \oplus \chi_R,$$

with a lepton number 0 (which is stable as shown in table 1; phenomenology doesn't depend on the specific choice of lepton number). The multiplet $\chi$ consists of three fields

$$\chi \sim (\chi_1, \chi_2, \chi_3),$$

the lightest of which becomes the DM. We call it minimal Flavored Dark Matter (mFDM). For simplicity, we work in the basis where the charged lepton mass matrix is diagonal so that $Y_l = \text{Diag}(y_e, y_\mu, y_\tau)$. The Lagrangian can be written while respecting MFV as:

$$\mathcal{L}_{\text{tot}} = \mathcal{L}_{\text{SM}} + \mathcal{L}_\chi, \tag{13}$$

where $\mathcal{L}_{SM}$ contains the SM interactions and

$$\mathcal{L}_\chi = \bar{\chi}\left(i\slashed{\partial} - m_\chi Y_l\right)\chi + \mathcal{L}_\chi^{eff} \tag{14}$$

$$= \bar{\chi}_1\left(i\slashed{\partial} - m_\chi y_e\right)\chi_1 + \bar{\chi}_2\left(i\slashed{\partial} - m_\chi y_\mu\right)\chi_2 + \bar{\chi}_3\left(i\slashed{\partial} - m_\chi y_\tau\right)\chi_3 + \mathcal{L}_\chi^{eff}, \tag{15}$$

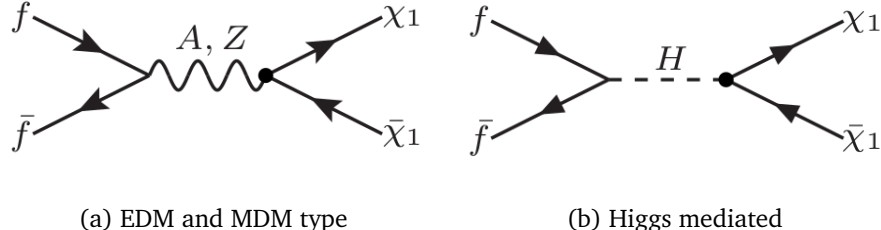

(a) EDM and MDM type        (b) Higgs mediated

Figure 1: The $2 \to 2$ annihilation channels relevant for dark matter freeze-in. $A, Z, H$ stand for the photon, Z and the Higgs fields. The blob represents the effective operator. Additional diagrams with the W bosons initial states are not shown.

where, $m_\chi$ is the only free parameter (dimension 1) and the mass ratios are fixed by MFV as:

$$\{m_{\chi_1}, m_{\chi_2}, m_{\chi_3}\} = m_\chi \{y_e, y_\mu, y_\tau\}. \tag{16}$$

Hence, $\chi_1$ becomes the lightest particle, the mFDM. Its interaction with SM is extracted from the lowest dimension effective operators:

$$\mathcal{L}_\chi^{eff} \supset \frac{1}{\Lambda_{MFV}} \underbrace{\left(\bar{\chi}_L \sigma_{\mu\nu} Y_l \chi_R\right) B^{\mu\nu}}_{MDM} + \frac{i}{\Lambda_{MFV}} \underbrace{\left(\bar{\chi}_L \sigma_{\mu\nu} \gamma_5 Y_l \chi_R\right) B^{\mu\nu}}_{EDM} + \frac{1}{\Lambda_{MFV}} \underbrace{\left(\bar{\chi}_L Y_l \chi_R\right) H^\dagger H}_{H-mediated}, \tag{17}$$

where EDM and MDM refer to the electric and magnetic dipole moment operators, respectively. The lightest particle, $\chi_1$, also the has the smallest couplings. In the following, we study the relic density and direct detection of $\chi_1$.

## 3.1 Relic Density

The freeze-in production mechanism assumes that DM had a negligible number density to begin with, and the observed relic density [41] is accumulated slowly from the annihilation or decay of SM particles to DM particles [11–13, 42]. The lowest dimension operators in eq. (17) give rise to two types of productions. The first two terms are dimension 5 operators and lead to UV freeze-in [43], see fig. (1a), with sensitivity to the reheating temperature $T_{RH}$. While the last term of eq. (17) gives also a renormalizable term after EWSB

$$\mathcal{L}_{\chi_1}^{eff} \supset \frac{vev}{\Lambda_{MFV}} y_e h \bar{\chi}_1 \chi_1,$$

giving rise to IR freeze-in [12], see fig. (1b) that is independent of the reheating temperature, albeit still dependent on the effective scale $\Lambda_{MFV}$. To compute the relic density one needs to solve the Boltzmann equations with appropriate initial conditions. In the present context, the Boltzmann equations are best expressed in terms of the yield, $Y \equiv n/\hat{s}$, the number density per co-moving volume, with $\hat{s}$ the entropy density. The collision term takes the form [44, 45]:

$$\frac{dY}{dT} = -\left(\frac{45}{\pi}\right)^{3/2} \frac{M_{Pl}}{4\pi^2 g_\star^s \sqrt{g_\star}} \left(1 + \frac{1}{3} \frac{d \ln g_\star}{d \ln T}\right) \frac{R(T)}{T^6}, \tag{18}$$

with the rate $R(T)$ given as:

$$R(T) = \sum_{i \in f, b} \frac{N_c^i}{(2\pi)^6 2^5} T \int_{s_{min}}^\infty ds \sqrt{s} \sqrt{1 - \frac{4m_i^2}{s}} \sqrt{1 - \frac{4m_{\chi_1}^2}{s}} K_1\left(\frac{\sqrt{s}}{T}\right) \int d\Omega_3^\star \sum |\mathcal{M}|_i^2. \tag{19}$$

The total rate is a sum of the contributions from each SM particle $i$ (fermion $f$ or boson $b$) with mass $m_i$, electromagnetic charge $q_i$, and color d.o.f. $N_c^i$. Here, $M_{Pl}$ is the Planck mass, $g_*$ and $g_*^s$ are the total number of effectively massless degrees of freedom contributing to energy and entropy densities, respectively, $s$ is the centre of mass energy with a minimum value of $s_{min} \equiv max(4m_i^2, 4m_{\chi_1}^2)$ for the annihilation process and $K_1$ is the modified Bessel function of the second kind. The squared amplitude is *summed* over initial and final spin states and after integration over solid angle between initial and final momenta, gives:

$$\int d\Omega_3^\star \sum |\mathcal{M}|_f^2 = \begin{cases} \frac{32\pi}{3s} c_W^2 e^2 q_f^2 \frac{y_e^2}{\Lambda_{MFV}^2}(s - 4m_{\chi_1}^2)(s + 2m_f^2), & \text{for EDM photon} \\ \frac{32\pi}{3s} c_W^2 e^2 q_f^2 \frac{y_e^2}{\Lambda_{MFV}^2}(s + 8m_{\chi_1}^2)(s + 2m_f^2), & \text{for MDM photon} \\ \frac{16\pi m_f^2}{((s-m_h^2)^2 + \Gamma_h^2 m_h^2)} \frac{y_e^2}{\Lambda_{MFV}^2}(s - 4m_{\chi_1}^2)(s - 4m_f^2), & \text{for Higgs mediated} \end{cases} \tag{20}$$

with, $c_W$ the cosine of Weinberg angle, $e$ the electromagnetic gauge coupling, $m_h$ the Higgs mass, $\Gamma_h$ the Higgs decay width and $y_e$ the electron Yukawa. The expressions for the $Z$ mediated production fig. (1a), are similar to those of the photon, with $c_W^2/s$ replaced by $s_W^2 s/((s - m_Z^2)^2 + \Gamma_Z^2 m_Z^2)$. It should be noted that, amplitudes involving $W$ bosons in the initial state are also present. The full computation takes in to account all the contributions.

The total relic density for $\chi_1$ is then given by:

$$\Omega h^2 = 2\frac{Y(T_0)s(T_0)m_\chi h^2}{\rho_{crit}} = 2\,m_{\chi_1}Y(T_0)\frac{2.9 \times 10^9\,\mathrm{m}^{-3}}{10.5\,\mathrm{GeV\,m}^{-3}}, \tag{21}$$

where the factor of 2 accounts for the anti-particles, $\rho_{crit}$ is the critical density, $Y(T_0)$ is the yield at temperature $T_0$ of the SM bath at the time of observation, and is calculated from eq. (18) by integrating over temperature from $T_0$ to $T_{RH}$, keeping in mind that $Y(T_{RH}) = 0$ from the assumption of negligible initial DM number density.

We verify the relic density obtained from micrOMEGAs 5.0 [46] against our calculations given above, for fermionic channels of production. And thereby proceed by using the code, which takes into account all $2 \rightarrow 2$ processes with SM gauge bosons and fermions, for calculating the freeze-in relic density. Note that the $2 \rightarrow 2$ Higgs/Z mediated channel also accounts for the Higgs/Z decay contributions when the mediator goes on-shell (see ref. [46]).

In the present work we restrict ourselves to masses of DM between a GeV to 100's of GeV, which is of interest at the liquid xenon/argon direct detection experiments that give the leading limit (Sec. 3.2). Additionally, we limit the parameter space to values where $\chi_2$, $\chi_3$ production is suppressed, i.e. $m_{\chi_2} >> T_{RH}$, so that the lightest particle $\chi_1$ with the coupling proportional to the electron Yukawa $y_e$ forms the majority of the relic density. Thus, $\chi_2$ relic density is less than 1% of the observed relic density. The relic density of $\chi_3$ is further suppressed. Given the Lagrangian in eq. (17), $\chi_2$ and $\chi_3$ are stable particles, and given their negligible abundances, they do not have any cosmological significance.

For far lower masses of DM, $\sim \mathcal{O}(\text{keV})$-$\mathcal{O}(\text{MeV})$, the role of the plasmon for our case has not been studied in great detail in literature and will be expanded upon in an upcoming publication [47].

The squared amplitudes from the dipole and Higgs-mediated operators with fermionic contributions to the relic density are shown in eq. (20). In order to understand the interplay between the "IR" and "UV" production channels of DM, we show in fig. (2) the relative contributions of the different channels. In fig. (2a), $T_{RH}$ is 5 GeV and the Higgs number density is Boltzmann suppressed making the magnetic dipole operator the dominant production channel. As the $T_{RH}$ increases to 50 GeV, the Higgs portal production becomes dominant when $T_{RH} \gtrsim m_h > m_{\chi_1}/2$, while the magnetic dipole moment operator starts dominating for larger $m_{\chi_1}$, as can be seen in fig. (2b). Also, we see that for $T_{RH} = 50$ GeV, the condition $m_{\chi_2} >> T_{RH}$

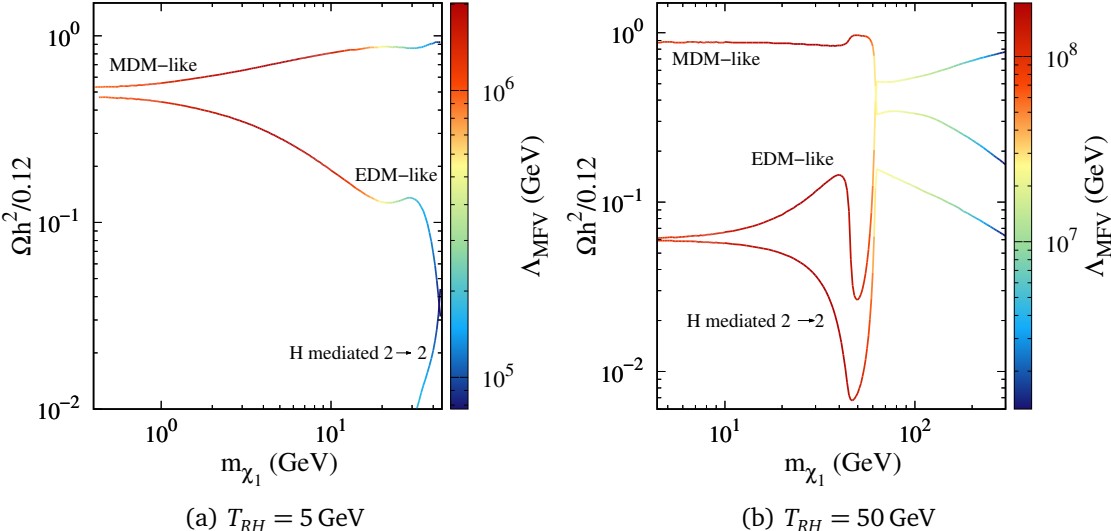

(a) $T_{RH} = 5\,\text{GeV}$        (b) $T_{RH} = 50\,\text{GeV}$

Figure 2: Fractional contribution of different annihilation channels to the total relic density, with $\Lambda_{MFV}$ (shown in color) chosen to reproduce the observed relic density for each DM mass.

sets the lower bound of $m_{\chi_1} \gtrsim 2.5\,\text{GeV}$. For larger still reheating temperatures, the pattern continues similarly. We also briefly note that although we have an effective photon interaction, the plasmon production of DM (see refs. [47–50]) is subdominant, owing to the more massive DM we consider here. For figs. (2), we have chosen $\Lambda_{MFV}$ corresponding to each DM mass such that each DM production channel adds up to give the total relic density of $\Omega h^2 = 0.12$.

## 3.2 Direct Detection

Freeze-in generated DM is notoriously difficult to test experimentally, owing to the very small couplings with SM particles (see ref. [51] for a review). On the other hand, light mediators with masses less than the exchanged momentum can increase the elastic scattering cross-section and the scattering rates at direct detection experiments [52]. The dipole operator interactions from eq. (17) of the mFDM consist of an elastic scattering process mediated by the photon whose Feynman diagram is shown in fig. (3). For masses of mFDM in the GeV range, observations from electron scattering do not probe the parameter space of interest [53, 54], even if one considers variations in the DM velocity distribution [55–57]. Thus, in the following, we discuss direct detection constraints on the model from nuclear scatterings.

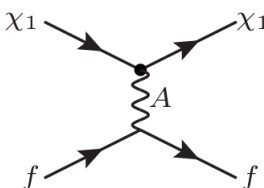

Figure 3: Diagram relevant to direct detection sourced from the dipole like operators. $A$ stands for the photon field. The blob represents the effective operator.

Direct detection of DM with dipolar interactions has garnered significant attention in literature

[58–76]. The differential rate of events, per unit time per unit mass, is given as [77–82]:

$$\frac{dR}{dE_R} = \frac{1}{m_N}\frac{\rho_{DM}}{m_{\chi_1}}\int_{v_{min}(E_R)}^{v_{esc}} dv\, f_\odot(v)\frac{d\sigma_N}{dE_R}(v, E_R)\, v\,, \tag{22}$$

where R is the rate, $E_R$ the recoil energy, $v$ the relative velocity between the DM particle and the nucleus, and $m_N$ the mass of the target nucleus for a given experiment. The local DM energy density, $\rho_{DM}$ is taken to be $0.3\,\text{GeV/cm}^3$ [83–86] and $f_\odot(v)$ is the velocity distribution of DM in Sun's rest frame [87, 88] given by:

$$f_\odot(v) = \frac{2\pi}{N_0}v^2\frac{v_0^2}{2vv_\odot}\left(\exp\left(-\frac{(v-v_\odot)^2}{v_0^2}\right) - \exp\left(-\frac{(v+v_\odot)^2}{v_0^2}\right)\right), \tag{23}$$

$$\text{with}\quad N_0 = \frac{\pi v_0^3}{2}\left[\sqrt{\pi}\left(\text{Erf}\left(\frac{v_{esc}-v_\odot}{v_0}\right) + \text{Erf}\left(\frac{v_{esc}+v_\odot}{v_0}\right)\right)\right.$$
$$\left. - \frac{v_0}{v_\odot}\left(\exp\left(-\frac{(v_{esc}-v_\odot)^2}{v_0^2}\right) - \exp\left(-\frac{(v_{esc}+v_\odot)^2}{v_0^2}\right)\right)\right], \tag{24}$$

the normalization. Here, $v_\odot$ is the velocity of Sun in the Galactic reference frame; the most probable velocity of DM is $v_0 = 220\,\text{km/s}$, and $v_{esc} = 540\,\text{km/s}$ is the local Galactic escape velocity [89]. In eq. (22), $v$ is integrated over, from a minimum velocity of $v_{min}(E_R) = \frac{1}{m_{red}}\sqrt{\frac{m_N E_R}{2}}$, which is necessary to produce a recoil energy of $E_R$, to a maximum velocity of $v_{esc}$. The reduced mass of the DM and the target nucleus is given by $m_{red} \equiv m_N m_{\chi_1}/(m_N + m_{\chi_1})$.

And finally, the differential cross section is given as [81]:

$$\frac{d\sigma_N}{dE_R} = \frac{1}{32\pi m_N m_{\chi_1}^2}\frac{1}{v^2}\overline{|\mathcal{M}|^2}\left|F(E_R)\right|^2\,, \tag{25}$$

with $\mathcal{M}$ the amplitude for the DM-nucleus(N) elastic scattering where-in the nucleus is treated as a point-like particle. $F(E_R)$ is the nuclear form factor that accounts for the finite size of the nucleus. It is a function of the momentum exchange $q$ ($E_R \equiv q^2/2m_N$), which could significantly differ from one, for large $q$.

For mFDM, the squared amplitudes in the non-relativistic limit, for scattering across a target nucleus with charge $Z$ and magnetic dipole moment $\mu_{Z,N}$ are:

$$\overline{|\mathcal{M}|^2} = \frac{1}{4}|\mathcal{M}|^2 \simeq \begin{cases} 8Z^2 e^2 c_W^2 y_e^2 m_{\chi_1}^2\frac{m_N}{E_R\Lambda_{MFV}^2}\,, & \text{for EDM} \\ 8Z^2 e^2 c_W^2 y_e^2 m_{\chi_1}^2\frac{m_N v^2}{E_R\Lambda_{MFV}^2}\left(1 + \frac{E_R}{2m_N v^2} - \frac{E_R}{m_{\chi_1}v^2}\right)\,, & \text{for MDM dipole-charge} \\ 16\,c_W^2 y_e^2 \mu_{Z,N}^2 m_{\chi_1}^2 m_N^2\frac{(v^2+2)}{\Lambda_{MFV}^2}\,, & \text{for MDM dipole-dipole} \end{cases} \tag{26}$$

and the corresponding differential cross sections become:

$$\frac{d\sigma_N}{dE_R} = \begin{cases} Z^2\alpha c_W^2 y_e^2\frac{1}{\Lambda_{MFV}^2}\frac{1}{v^2 E_R}\left|F_E(E_R)\right|^2\,, & \text{for EDM} \\ Z^2\alpha c_W^2 y_e^2\frac{1}{\Lambda_{MFV}^2}\frac{1}{E_R}\left(1 + \frac{E_R}{2m_N v^2} - \frac{E_R}{m_{\chi_1}v^2}\right)\left|F_E(E_R)\right|^2\,, & \text{for MDM dipole-charge} \\ \frac{\alpha c_W^2 y_e^2}{E_R\Lambda_{MFV}^2}\left(\mu_{Z,N}\Big/\frac{e}{2m_n}\right)^2\frac{m_N E_R}{m_n^2 v^2}\left|F_M(E_R)\right|^2\,, & \text{for MDM dipole-dipole} \end{cases} \tag{27}$$

where $F_M$ is the magnetic nuclear form factor normalized to $F_M(0) = 1$, and $F_E$ is the nuclear charge form factor normalized to $F_E(0) = 1$. For the numerical analysis we take $F_E$ and $F_M$ to be approximately equal to the Helm form factor, $F(E_R) = \frac{3j_1(qr_n)}{qr_n}e^{-(qs)^2/2}\big|_{q=\sqrt{2m_N E_R}}$, where $j_1$ is the first spherical Bessel function. For a heavy nuclei, a good approximation is given by

$r_n \approx 1.14 A^{1/3}$ fm and $s \approx 0.9$ fm, for the radial and surface thickness parameters, respectively [59, 81, 90], where $A$ denotes the atomic weight of the target nucleus.

We see that the EDM contribution scales as $1/(v^2 E_R)$, and gives the largest contribution in the non-relativistic limit of small $v$ and $E_R$. This is why the direct detection limits from EDM are the most constraining and is the only one that shows up in fig. (4). To obtain this constraint, we consider the XENON1T experiment [89], which gives the strongest constraint for DM with masses greater than a few GeV. Current generation liquid argon experiments are not sensitive to the model studied in this paper [91, 92], however, future liquid argon detectors like DarkSide-20k [93] and ARGO [94, 95] will probe these models. For concreteness, we only focus on xenon-based experiments.

One must then calculate the total number of scattering events, as the dipolar mediated elastic scattering cannot be approximated to a contact interaction (with bounds given as $\sigma_{SI}$ vs. $m_\chi$). This is given as:

$$N = 0.475 \, M_T \times \Delta t \times \int_{E_R^{min}}^{E_R^{max}} dE_R \, \epsilon(E_R) \frac{dR}{dE_R} \,, \tag{28}$$

where $M_T$ is the total mass of the target liquid xenon, $\Delta t$ is the time period over which the data is taken, $\epsilon(E_R)$ is the efficiency taken from fig. 1 of ref. [89] and the factor of 0.475 accounts for the reference region, fig. 3 of ref. [89]. Here the integration over the recoil energy is from $E_R^{min} = 4.9$ keV to $E_R^{max} = 40.9$ keV. From table 1 of ref. [89], we demand for the total number of events to be less than 1.7 over 1 ton×year, deriving the bounds on $\Lambda_{MFV}$ for a given $m_{\chi_1}$. In fig. (4a), we show the combined results for relic density as well as direct detection for a range of masses and $\Lambda_{MFV}$. We have shown the contours for $\Omega h^2 = 0.12$ for reheating temperatures $T_{RH} = 5$ (red), 10 (magenta), 20 (orange) and 50 (blue) GeV. As expected, larger $T_{RH}$ mandates smaller coupling, and hence larger $\Lambda_{MFV}$, at the behest of the UV production. For each $T_{RH}$, the cut-off/edge in low $m_{\chi_1}$ is owing to the condition of $m_{\chi_2} \gg T_{RH}$ while the cut-off/edge at large $m_{\chi_1}$ values is from requiring $\Lambda_{MFV} > m_{\chi_3}$ (this condition sets in even before the Boltzmann suppression makes it infeasible to produce a DM much heavier than the $T_{RH}$). The downward dip in each case is because of a decreasing $\Lambda_{MFV}$ (increasing effective coupling) required to compensate for the Boltzmann suppression for $m_{\chi_1} > T_{RH}$. We also observe a sharp dip at $m_{\chi_1} \sim m_h/2$ that is explained by the Higgs production going off-shell, requiring larger couplings (smaller $\Lambda_{MFV}$) to reproduce the observed relic density.

Also shown in fig. (4a) are the limits from XENON1T data (yellow region) which is not yet sensitive to the region of interest of this effective theory model. It should be noted that relaxing the constraint of $m_{\chi_3} \lesssim \Lambda_{MFV}$, by utilising the unknown $\mathcal{O}(1)$ factors in the coefficients of these operators, would put the model under the scrutiny of XENON1T data. We also present the projected bounds from a 2000 ton×year run of DARWIN [96] and a $\sim$15 ton×year run of LZ [97], which would probe the parameter space relevant for a $T_{RH}$ of a few GeV. Both these constraints arise from the EDM type interaction that dominates, as discussed above, at the low velocities of these experiments. A comment is in order regarding the effective theory conditions we have imposed $viz$, $m_{\chi_3} \lesssim \Lambda_{MFV}$. Removing this condition makes larger parameter space accessible at the Direct Detection experiments. This condition can be significantly relaxed by considering a smaller flavor group $SU(2)$ instead of $SU(3)$. In this case, only the first two generations are considered to have non-trivial representations under the flavor group [98, 99], when the flavor group is broken from $SU(3)$ to $SU(2)$. Assuming chiral doublet representations for the DM, in a similar fashion to the triplet case, we plot the current bounds from XENON1T in fig. (4b). Note that we have imposed the relevant effective field theory constraints, $m_{\chi_2} < \Lambda_{MFV}$ here too. As can be seen, a significant region of the parameter space is already ruled out from XENON1T data with LZ and DARWIN projected to probe a much larger region of this model. There exist FIMP models that derive constraints from indirect detection

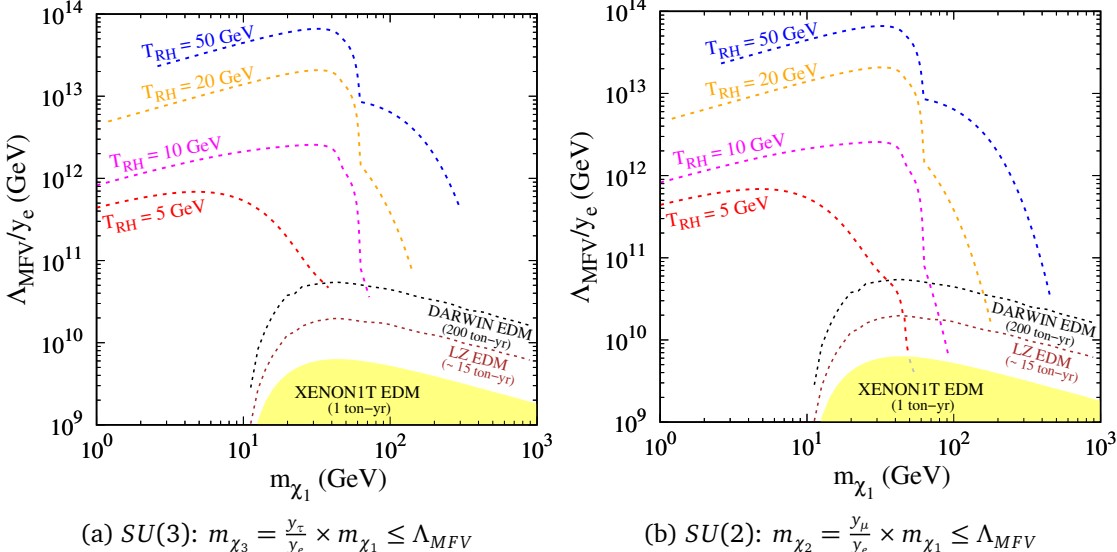

Figure 4: Results from relic density constraints and direct detection experiments: the contours correspond to the parameter space exactly reproducing observed relic density, for given reheating temperatures. The yellow shaded region is ruled out by current bounds from XENON1T, and the future projections of constraints from DARWIN and LZ are also shown.

experiments [14, 100], CMB and astrophysics experiments [48, 50, 101–104], and at colliders [105, 106]. However, we estimate that these constraints do not probe the models studied in this work [107–111]. Neutron star constraints [112] are not able to probe these models either because they probe much heavier masses or because the capture rate is suppressed due to the large velocity of DM particles near the neutron star surface.

# 4 Conclusions and outlook

The charged lepton flavor group provides a natural framework for freeze-in DM. In particular, representations that are chiral under the charged lepton flavor group provide the coupling of the right order to satisfy the relic density. The dimension 5 operators of the effective theories contain magnetic and electric dipole moment operators in addition to the Higgs portal terms. These play an important role, especially in testing this model, as the elastic scattering cross section from the electric dipole operator is significantly enhanced at low momenta. In the relic density, for low reheating scenarios, the magnetic dipole operator dominates whereas for higher reheating temperatures, $T_{RH} \gtrsim 20$ GeV, the Higgs portal dominates.

The phenomenological analysis in this work has been restricted to the dimension 5 effective field theory. There could be higher dimensional operators at dimension 6 etc., and it would be interesting to study the implications of those operators, at least for specific representations. The operators can be classified in terms of two types: those with Yukawa insertions as dictated by MFV at the leading order and those with Yukawa insertions as dictated by MFV at the sub-leading order. For example, some vector-vector four fermion operators would fall in the second category. In that case a full numerical analysis and perhaps some additional assumptions (perhaps on Z' masses etc) might be needed to make connection with the phenomenology. The wide range of prospects makes a detailed study outside the scope of this work.

While not a necessary condition, assuming total lepton number conservation assures the stability of these DM particles. This naturally fits in with models with Dirac neutrinos [113] and Dirac neutrinogenesis [114] and can lead to construction of models with viable neutrino sector. It is interesting to note that right handed neutrinos interacting solely via weak interactions are well within the bounds from $N_{eff}$ as long as $m_\nu < 100$ keV [115, 116]. The $G_{LF}$ chosen in the paper including lepton number conservation would allow for operators which might be suppressed by the neutrino mass operator in most cases depending on the particle spectrum and the representation allowed. This would require a separate investigation which has not been included in the paper.

On the other hand, relations with tiny lepton number violation for Majorana neutrinos and DM stability would be interesting to study. Overall, flavored DM seems to be an ideal candidate where one can expect a "FIMP" miracle to take place.

## Acknowledgements

We thank Alejandro Ibarra and Mu-Chun Chen for interesting discussions. We also thank Jae Hyeok Chang, Xiaoyong Chu and Gaurav Tomar for correspondence and helpful discussions. G.D. was supported in part by MIUR under Project No. 2015P5SBHT and by the INFN research initiative ENP. G.D. thanks "Satish Dhawan Visiting Chair Professorship" at the Indian Institute of Science. We thank IoE funds of IISc for support. S.K.V. is supported by the project "Nature of New Physics" by the Department of Science and Technology, Govt. of India.

## A  Amplitudes for relic density and direct detection

Amplitudes for $2 \to 2$ annihilation production of relic density:

$$
\begin{aligned}
\mathcal{M}_{EDM} &= \frac{iq_f\, ec_W y_e}{2\Lambda_{MFV}} \frac{(p_1+p_2)^\alpha}{s} \Big(\bar{u}(p_3, m_{\chi_1}).(\gamma_\alpha\gamma_\mu - \gamma_\mu\gamma_\alpha).\gamma_5.v(p_4, m_{\chi_1})\Big) \\
&\quad \times\Big(\bar{v}(p_2, m_f)\gamma_\mu u(p_1, m_f)\Big), \quad (29) \\
\mathcal{M}_{MDM} &= \frac{q_f\, ec_W y_e}{2\Lambda_{MFV}} \frac{(p_1+p_2)^\alpha}{s} \Big(\bar{u}(p_3, m_{\chi_1}).(\gamma_\alpha\gamma_\mu - \gamma_\mu\gamma_\alpha).v(p_4, m_{\chi_1})\Big) \\
&\quad \times\Big(\bar{v}(p_2, m_f)\gamma_\mu u(p_1, m_f)\Big), \quad (30) \\
\mathcal{M}_H &= \frac{m_f y_e}{\Lambda_{MFV}} \Big(\bar{u}(p_3, m_{\chi_1}).v(p_4, m_{\chi_1})\Big)\Big(\bar{v}(p_2, m_f).u(p_1, m_f)\Big). \quad (31)
\end{aligned}
$$

Amplitudes for direct detection calculations:

$$
\begin{aligned}
\text{EDM: } \mathcal{M}_{EDM} &= Ze\frac{y_e c_W}{2\Lambda_{MFV}} \frac{1}{(p_3-p_1)^2}\Big(\bar{u}(p_4, m_N)\gamma^\nu u(p_2, m_N)\Big) \\
&\quad \times\Big(\bar{u}(p_3, m_{\chi_1})(\gamma^\mu\gamma^\nu - \gamma^\nu\gamma^\mu)\gamma_5 u(p_1, m_{\chi_1})\Big)(p_3-p_1)^\mu, \quad (32) \\
\text{MDM dipole charge: } \mathcal{M}^{SI}_{MDM} &= Ze\frac{y_e c_W}{2\Lambda_{MFV}} \frac{i}{(p_3-p_1)^2}\Big(\bar{u}(p_4, m_N)\gamma^\nu u(p_2, m_N)\Big) \\
&\quad \times\Big(\bar{u}(p_3, m_{\chi_1})(\gamma^\mu\gamma^\nu - \gamma^\nu\gamma^\mu) u(p_1, m_{\chi_1})\Big)(p_3-p_1)^\mu, \quad (33) \\
\text{MDM dipole-dipole: } \mathcal{M}^{SD}_{MDM} &= \frac{y_e c_W}{2\Lambda_{MFV}} \frac{i}{q^2} \frac{\mu_{Z,N}}{2}\Big(\bar{u}(p_4, m_N)(\gamma^\nu\gamma^\alpha - \gamma^\alpha\gamma^\nu) u(p_2, m_N)\Big)(p_2-p_4)^\alpha \\
&\quad \times\Big(\bar{u}(p_3, m_{\chi_1})(\gamma^\mu\gamma^\nu - \gamma^\nu\gamma^\mu) u(p_1, m_{\chi_1})\Big)(p_3-p_1)^\mu. \quad (34)
\end{aligned}
$$

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
