# Peer review of "Freezing In with Lepton Flavored Fermions"

_SciPost Physics, doi:SciPost Phys. 11, 006 (2021)_

## Round 1 · Referee Report · Anonymous (Referee 1) · 2021-5-3

Report

The authors study lepton flavoured fermions as freeze-in dark matter candidates. These models are attractive because dark matter stability can follow automatically from lepton number conservation. And freeze-in appears natural as a consequence of the smallness of the electron Yukawa coupling. The authors choose a minimal model and analyse the relic abundance constraint as well as direct detection constraints on the model. The analysis is solid and the results appear reasonable, the paper certainly meets the acceptance criteria of the journal. There are a few minor points that could be discussed in more detail or improved in my opinion:

  1. I think it would be useful to show Lambda_MFV, e.g. in Fig. 2. This would give the readers a better idea how suppressed e.g. higher dimensional operators would be.

  2. For the relic density computation, I wonder if there could be a relevant contribution from Z decays to DM pairs, while it appears that only 2->2 scatterings were taken into account. Also WW-> chi_1 chi_1 scatterings should become relevant at higher reheating temperatures.

  3. What is the lifetime of the heavier DM components? These are typically more strongly coupled due to the larger Yukawas, so they might play a role in the cosmological history of the model.

  4. The caption of Fig. 4 is a bit short. It should be explained more clearly what the T_RH contours mean (I assume these are the relic density contours for the given T_RH value?)

  5. I would like to see a discussion of the dimension 6 Lagrangian. Some operators there might be allowed that could affect the relic density if Lambda_MFW is not too large. I wonder in particular if at dimension 6 one can write down operators which are invariant under G_LF that involve both leptons and dark matter particles, and which are not suppressed by insertions of Yukawa couplings.

  6. Also a longer discussion of neutrino masses would be nice. If Lepton number is imposed as exact global symmetry, then the neutrinos are Dirac, and the RH neutrinos should also transform as multiplets of the Lepton flavour group. This could allow many more operators maybe even at the level of dimension 4 and 5. Furthermore additional constraints arise, since the RH neutrinos should not be thermalised, otherwise Neff constraints could be violated.

Requested changes

See report.

  • validity: high
  • significance: good
  • originality: good
  • clarity: high
  • formatting: excellent
  • grammar: excellent

Author:  Shiuli Chatterjee  on 2021-06-06  [id 1490]

(in reply to Report 1 on 2021-05-03)
Category:
answer to question

We thank the referee for finding the work acceptable for publication in SciPost and for the positive comments. We have carefully considered all the comments made by the referee and will make the corresponding changes in the next version of the manuscript. With these changes, we hope that the referee will recommend the paper for publication.

Referee comments: 1. I think it would be useful to show Lambda_MFV, e.g. in Fig. 2. This would give the readers a better idea how suppressed e.g. higher dimensional operators would be. Our response: We have now made a new version of Fig 2, which has the information of Lambda_{MFV}. They are attached with this message.

Referee comments: 2. For the relic density computation, I wonder if there could be a relevant contribution from Z decays to DM pairs, while it appears that only 2->2 scatterings were taken into account. Also WW-> chi_1 chi_1 scatterings should become relevant at higher reheating temperatures. Our response: We have given only the photon and Higgs mediated cross sections for illustrative purposes and to display the analytical expressions. But for the full calculation for relic density, we use Micromegas, after verifying that the relic densities match with our analytical calculations. In Micromegas, all 2→2 processes are accounted for properly, including W^+ W^- \rightarrow DM DM. As for decay channels, a Z decay process is the same as a SM SM \rightarrow Z \rightarrow DM DM with Z on-shell (i.e. the 2 \rightarrow 2 processes mediated by an on-shell Z). And therefore the 2 → 2 processes take all decays into account. These are briefly mentioned on page 10; and we will expand upon the relic density calculation in text upon re-submission.

Referee comments: 3. What is the lifetime of the heavier DM components? These are typically more strongly coupled due to the larger Yukawas, so they might play a role in the cosmological history of the model. Our response: We considered a flavored triplet (\chi_1,\chi_2,\chi_3), with \chi_1 being the DM. We think the referee is pointing towards \chi_2 and \chi_3 when talking about “the heavier DM components”. We constrain the parameter space to values where \chi_2, \chi_3 production is suppressed to ensure that the electron Yukawa sets the relic density, and not the larger muon or tau Yukawas. Thus in the chosen the parameter space \chi_2 relic density is less than 1% of the observed relic density. And \chi_3 relic density is further suppressed. The suppression is kinematic, as m_{\chi_2}, m_{\chi_3} >> T_{RH}. We also note that \chi_2 and \chi_3 are stable particles, given our Lagrangian, and given their negligible abundances, they do not have any cosmological significance. We will expand the paragraph following eq. (22) in section III-A of the paper, upon re-submission, so as to make this point explicit.

Referee comments: 4. The caption of Fig. 4 is a bit short. It should be explained more clearly what the T_RH contours mean (I assume these are the relic density contours for the given T_RH value?) Our response: Yes, the contours correspond to the parameter space exactly reproducing observed relic density, for given reheating temperature. The details for this are given in section III-A, following eq. (22), and we will expand the caption of fig. 4 to reflect this.

Referee comments: 5. I would like to see a discussion of the dimension 6 Lagrangian. Some operators there might be allowed that could affect the relic density if Lambda_MFW is not too large. I wonder in particular if at dimension 6 one can write down operators which are invariant under G_LF that involve both leptons and dark matter particles, and which are not suppressed by insertions of Yukawa couplings. Our response: The full dimension 6 effective Lagrangian would contain several operators which might have to be dealt with on a case-by-case basis, as we briefly note in the conclusions section. The operators can be classified in terms of two types: those with Yukawa insertions as dictated by MFV at the leading order and those with Yukawa insertions as dictated by MFV at the subleading order. For example, some vector-vector four fermion operators would fall in the second category. In that case a full numerical analysis and perhaps some additional assumptions (perhaps on Z’ masses etc) might be needed to make connection with the phenomenology. Given the wide range of prospects possible, we refrained from making further comments. We will expand upon the statement in the conclusions section in text upon re-submission.

Referee comments: 6. Also a longer discussion of neutrino masses would be nice. If Lepton number is imposed as exact global symmetry, then the neutrinos are Dirac, and the RH neutrinos should also transform as multiplets of the Lepton flavour group. This could allow many more operators maybe even at the level of dimension 4 and 5. Furthermore additional constraints arise, since the RH neutrinos should not be thermalised, otherwise Neff constraints could be violated. Our response: We agree with the referee that the neutrino sector is quite interesting, as we have briefly noted in the concluding section. The G_{LF} chosen in the paper including lepton number conservation would allow for operators which might be suppressed by the neutrino mass operator in most cases depending on the particle spectrum and the representation allowed. This would require a separate investigation which has not been included in the paper. But, this is an interesting direction of investigation which we are currently pursuing. For right handed neutrinos, thermalization with SM does not occur as long as m_\nu < 100 keV and weak interactions are the only interactions producing neutrinos (as discussed by Blinov et al, hep-ph/1905.02727, and Dolgov, hep-ph/0202122). This can be ensured for specific models of Dirac neutrinos.

Attachment:

LMFV_figs2.pdf

Tim Tait  on 2021-06-08  [id 1496]

(in reply to Shiuli Chatterjee on 2021-06-06 [id 1490])
Category:
answer to question

We look forward to receiving your revised manuscript as described.

---

## Round 2 · Author Response

We thank the referee for finding the work acceptable for publication in SciPost and for the positive comments. We have carefully considered all the comments made by the referee and will make the corresponding changes in the next version of the manuscript. With these changes, we hope that the referee will recommend the paper for publication.
Referee comments:
“1. I think it would be useful to show Lambda_MFV, e.g. in Fig. 2. This would give the readers a better idea how suppressed e.g. higher dimensional operators would be.”
Our response:
We have now made a new version of Fig 2, which has the information of Lambda_{MFV}.
Referee comments:
“2. For the relic density computation, I wonder if there could be a relevant contribution from Z decays to DM pairs, while it appears that only 2->2 scatterings were taken into account. Also WW-> chi_1 chi_1 scatterings should become relevant at higher reheating temperatures.”
Our response:
We have given only the photon and Higgs mediated cross sections for illustrative purposes and to display the analytical expressions. But for the full calculation for relic density, we use Micromegas, after verifying that the relic densities match with our analytical calculations. In Micromegas, all 2→2 processes are accounted for properly, including W^+ W^- \rightarrow DM DM. As for decay channels, a Z decay process is the same as a SM SM \rightarrow Z \rightarrow DM DM with Z on-shell (i.e. the 2 \rightarrow 2 processes mediated by an on-shell Z). And therefore the 2 → 2 processes take all decays into account.
These are briefly mentioned on page 10; and we will expand upon the relic density calculation in text upon re-submission.
Referee comments:
“3. What is the lifetime of the heavier DM components? These are typically more strongly coupled due to the larger Yukawas, so they might play a role in the cosmological history of the model.”
Our response:
We considered a flavoured triplet (\chi_1,\chi_2,\chi_3), with \chi_1 being the DM. We think the referee is pointing towards \chi_2 and \chi_3 when talking about “the heavier DM components”. We constrain the parameter space to values where \chi_2, \chi_3 production is suppressed to ensure that the electron Yukawa sets the relic density, and not the larger muon or tau Yukawas. Thus in the chosen the parameter space \chi_2 relic density is less than 1% of the observed relic density. And \chi_3 relic density is further suppressed. The suppression is kinematic, as m_{\chi_2}, m_{\chi_3} >> T_{RH}.
We also note that \chi_2 and \chi_3 are stable particles, given our Lagrangian, and given their negligible abundances, they do not have any cosmological significance. We will expand the paragraph following eq. (22) in section III-A of the paper, upon re-submission, so as to make this point explicit.
Referee comments:
“4. The caption of Fig. 4 is a bit short. It should be explained more clearly what the T_RH contours mean (I assume these are the relic density contours for the given T_RH value?)”
Our response:
Yes, the contours correspond to the parameter space exactly reproducing observed relic density, for given reheating temperature. The details for this are given in section III-A, following eq. (22), and we will expand the caption of fig. 4 to reflect this.
Referee comments:
“5. I would like to see a discussion of the dimension 6 Lagrangian. Some operators there might be allowed that could affect the relic density if Lambda_MFW is not too large. I wonder in particular if at dimension 6 one can write down operators which are invariant under G_LF that involve both leptons and dark matter particles, and which are not suppressed by insertions of Yukawa couplings.”
Our response:
The full dimension 6 effective Lagrangian would contain several operators which might have to be dealt with on a case-by-case basis, as we briefly note in the conclusions section.
The operators can be classified in terms of two types: those with Yukawa insertions as dictated by MFV at the leading order and those with Yukawa insertions as dictated by MFV at the subleading order. For example, some vector-vector four fermion operators would fall in the second category. In that case a full numerical analysis and perhaps some additional assumptions (perhaps on Z’ masses etc) might be needed to make connection with the phenomenology. Given the wide range of prospects possible, we refrained from making further comments. We will expand upon the statement in the conclusions section in text upon re-submission.
Referee comments:
"6. Also a longer discussion of neutrino masses would be nice. If Lepton number is imposed as exact global symmetry, then the neutrinos are Dirac, and the RH neutrinos should also transform as multiplets of the Lepton flavour group. This could allow many more operators maybe even at the level of dimension 4 and 5. Furthermore additional constraints arise, since the RH neutrinos should not be thermalised, otherwise Neff constraints could be violated."
Our response:
We agree with the referee that the neutrino sector is quite interesting, as we have briefly noted in the concluding section. The G_{LF} chosen in the paper including lepton number conservation would allow for operators which might be suppressed by the neutrino mass operator in most cases depending on the particle spectrum and the representation allowed. This would require a separate investigation which has not been included in the paper. But, this is an interesting direction of investigation which we are currently pursuing.
For right handed neutrinos, thermalization with SM does not occur as long as m_\nu < 100 keV and weak interactions are the only interactions producing neutrinos (as discussed by Blinov et al, hep-ph/1905.02727, and Dolgov, hep-ph/0202122). This can be ensured for specific models of Dirac neutrinos.
Referee comments:
“1. I think it would be useful to show Lambda_MFV, e.g. in Fig. 2. This would give the readers a better idea how suppressed e.g. higher dimensional operators would be.”
Our response:
We have now made a new version of Fig 2, which has the information of Lambda_{MFV}.
Referee comments:
“2. For the relic density computation, I wonder if there could be a relevant contribution from Z decays to DM pairs, while it appears that only 2->2 scatterings were taken into account. Also WW-> chi_1 chi_1 scatterings should become relevant at higher reheating temperatures.”
Our response:
We have given only the photon and Higgs mediated cross sections for illustrative purposes and to display the analytical expressions. But for the full calculation for relic density, we use Micromegas, after verifying that the relic densities match with our analytical calculations. In Micromegas, all 2→2 processes are accounted for properly, including W^+ W^- \rightarrow DM DM. As for decay channels, a Z decay process is the same as a SM SM \rightarrow Z \rightarrow DM DM with Z on-shell (i.e. the 2 \rightarrow 2 processes mediated by an on-shell Z). And therefore the 2 → 2 processes take all decays into account.
These are briefly mentioned on page 10; and we will expand upon the relic density calculation in text upon re-submission.
Referee comments:
“3. What is the lifetime of the heavier DM components? These are typically more strongly coupled due to the larger Yukawas, so they might play a role in the cosmological history of the model.”
Our response:
We considered a flavoured triplet (\chi_1,\chi_2,\chi_3), with \chi_1 being the DM. We think the referee is pointing towards \chi_2 and \chi_3 when talking about “the heavier DM components”. We constrain the parameter space to values where \chi_2, \chi_3 production is suppressed to ensure that the electron Yukawa sets the relic density, and not the larger muon or tau Yukawas. Thus in the chosen the parameter space \chi_2 relic density is less than 1% of the observed relic density. And \chi_3 relic density is further suppressed. The suppression is kinematic, as m_{\chi_2}, m_{\chi_3} >> T_{RH}.
We also note that \chi_2 and \chi_3 are stable particles, given our Lagrangian, and given their negligible abundances, they do not have any cosmological significance. We will expand the paragraph following eq. (22) in section III-A of the paper, upon re-submission, so as to make this point explicit.
Referee comments:
“4. The caption of Fig. 4 is a bit short. It should be explained more clearly what the T_RH contours mean (I assume these are the relic density contours for the given T_RH value?)”
Our response:
Yes, the contours correspond to the parameter space exactly reproducing observed relic density, for given reheating temperature. The details for this are given in section III-A, following eq. (22), and we will expand the caption of fig. 4 to reflect this.
Referee comments:
“5. I would like to see a discussion of the dimension 6 Lagrangian. Some operators there might be allowed that could affect the relic density if Lambda_MFW is not too large. I wonder in particular if at dimension 6 one can write down operators which are invariant under G_LF that involve both leptons and dark matter particles, and which are not suppressed by insertions of Yukawa couplings.”
Our response:
The full dimension 6 effective Lagrangian would contain several operators which might have to be dealt with on a case-by-case basis, as we briefly note in the conclusions section.
The operators can be classified in terms of two types: those with Yukawa insertions as dictated by MFV at the leading order and those with Yukawa insertions as dictated by MFV at the subleading order. For example, some vector-vector four fermion operators would fall in the second category. In that case a full numerical analysis and perhaps some additional assumptions (perhaps on Z’ masses etc) might be needed to make connection with the phenomenology. Given the wide range of prospects possible, we refrained from making further comments. We will expand upon the statement in the conclusions section in text upon re-submission.
Referee comments:
"6. Also a longer discussion of neutrino masses would be nice. If Lepton number is imposed as exact global symmetry, then the neutrinos are Dirac, and the RH neutrinos should also transform as multiplets of the Lepton flavour group. This could allow many more operators maybe even at the level of dimension 4 and 5. Furthermore additional constraints arise, since the RH neutrinos should not be thermalised, otherwise Neff constraints could be violated."
Our response:
We agree with the referee that the neutrino sector is quite interesting, as we have briefly noted in the concluding section. The G_{LF} chosen in the paper including lepton number conservation would allow for operators which might be suppressed by the neutrino mass operator in most cases depending on the particle spectrum and the representation allowed. This would require a separate investigation which has not been included in the paper. But, this is an interesting direction of investigation which we are currently pursuing.
For right handed neutrinos, thermalization with SM does not occur as long as m_\nu < 100 keV and weak interactions are the only interactions producing neutrinos (as discussed by Blinov et al, hep-ph/1905.02727, and Dolgov, hep-ph/0202122). This can be ensured for specific models of Dirac neutrinos.

---

## Round 2 · List of Changes

- Figs. 2 changed to reflect the scale \Lambda_{MFV}
- Improved discussion in section III-A to clarify that (a) heavier dark particles are not of cosmological significance (b) SM bosons in DM production are accounted for, in both annihilation and decay processes
- Caption for figs. 4 expanded
- Discussions added in section IV on viability of right handed neutrinos and of the scope of higher dimension operators
- Removed typos and added references

---

## Editorial Decision

published